# Calcium Signaling Silencing in Atrial Fibrillation: Implications for Atrial Sodium Homeostasis

**DOI:** 10.3390/ijms221910513

**Published:** 2021-09-29

**Authors:** Aaron D. Kaplan, Humberto C. Joca, Liron Boyman, Maura Greiser

**Affiliations:** 1Center for Biomedical Engineering and Technology, Department of Physiology, University of Maryland School of Medicine, Baltimore, MD 21201, USA; akaplan@som.umaryland.edu (A.D.K.); hjoca@som.umaryland.edu (H.C.J.); lboyman@som.umaryland.edu (L.B.); 2Division of Cardiovascular Medicine, Department of Medicine, University of Maryland School of Medicine, Baltimore, MD 21201, USA

**Keywords:** atrial fibrillation, calcium signaling silencing, Ca^2+^ handling, Na^+^ concentration, atrial myocytes, Na^+^ currents, ranolazine

## Abstract

Atrial fibrillation (AF) is the most common type of cardiac arrhythmia, affecting more than 33 million people worldwide. Despite important advances in therapy, AF’s incidence remains high, and treatment often results in recurrence of the arrhythmia. A better understanding of the cellular and molecular changes that (1) trigger AF and (2) occur after the onset of AF will help to identify novel therapeutic targets. Over the past 20 years, a large body of research has shown that intracellular Ca^2+^ handling is dramatically altered in AF. While some of these changes are arrhythmogenic, other changes counteract cellular arrhythmogenic mechanisms (Calcium Signaling Silencing). The intracellular Na^+^ concentration ([Na^+^])_i_ is a key regulator of intracellular Ca^2+^ handling in cardiac myocytes. Despite its importance in the regulation of intracellular Ca^2+^ handling, little is known about [Na^+^]_i_, its regulation, and how it might be changed in AF. Previous work suggests that there might be increases in the late component of the atrial Na^+^ current (I_Na,L_) in AF, suggesting that [Na^+^]_i_ levels might be high in AF. Indeed, a pharmacological blockade of I_Na,L_ has been suggested as a treatment for AF. Here, we review calcium signaling silencing and changes in intracellular Na^+^ homeostasis during AF. We summarize the proposed arrhythmogenic mechanisms associated with increases in I_Na,L_ during AF and discuss the evidence from clinical trials that have tested the pharmacological I_Na,L_ blocker ranolazine in the treatment of AF.

## 1. Introduction

Atrial fibrillation (AF) is the most common type of arrhythmia, affecting more than 33 million patients worldwide [1]. AF is a supraventricular arrhythmia with high and irregular atrial activation rates. Clinically, AF is characterized by a multitude of symptoms including fatigue, dyspnea, syncope, hypotension, and tachycardia-induced cardiomyopathy [2]. Further, AF provides a nidus for thromboembolism and stroke and is a serious source of morbidity and mortality [1]. The development of AF is a complex process involving an interplay between electrical and structural remodeling, autonomic imbalance, alterations in Ca^2+^ handling, abnormal metabolism, and genetic factors. Treatment for AF is limited by the suboptimal efficacy and toxicities of pharmaceuticals or radiofrequency ablation or surgery, as well as high rates of recurrence [3].

Many vital cell functions are regulated by the intracellular Ca^2+^ homeostasis in cardiac myocytes. Intracellular Ca^2+^ release initiates contraction, shapes the action potential, regulates transcription, and is involved in a myriad of intracellular signaling cascades [4,5]. AF leads to complex remodeling of the intracellular Ca^2+^ homeostasis in atrial myocytes. Changes in intracellular Ca^2+^ signaling during AF contribute to a reduction in contractile function (‘contractile remodeling’ [6]) and shortening of the atrial myocyte’s action potential (AP) (‘electrical remodeling’ [7,8]). Whether AF-induced changes in intracellular Ca^2+^ signaling also contribute to Ca^2+^-based arrhythmogenesis in AF remains controversial [9,10,11,12]. Previous work has identified unstable intracellular Ca^2+^ signaling in atrial myocytes from patients with AF that was compatible with intracellular arrhythmogenic mechanisms. At the same time, AF leads to ‘calcium signaling silencing’, an adaptive mechanism that counteracts intracellular Ca^2+^ overload [9,10,11,13].

In cardiac myocytes, the intracellular Na^+^ concentration ([Na^+^]_i_) is an important regulator of the intracellular Ca^2+^ concentration ([Ca^2+^]_i_) via the sarcolemmal Na^+^/Ca^2+^ exchanger (NCX), which exchanges 3 Na^+^ for 1 Ca^2+^ [14,15,16]. A high [Na^+^]_i_ concentration is known as a potent driver of Ca^2+^-based arrhythmogenesis in ventricular heart disease (i.e., heart failure) [17,18,19]. Studies have indicated that high [Na^+^]_i_ concentrations in atrial myocytes might contribute to Ca^2+^-based mechanisms of arrhythmia in AF [20,21]. Specifically, increases in the late component of the cardiac Na^+^ current (I_Na,L_) have been suggested to contribute to arrhythmogenesis in AF [20]. We and others have comprehensively reviewed changes in intracellular Ca^2+^ signaling in AF and how they relate to atrial arrhythmogenesis [9,22,23]. Here, we want to focus on the relationship between intracellular Ca^2+^ and Na^+^ homeostasis and its implications for arrhythmogenesis in AF. We briefly discuss altered Ca^2+^ handling in AF and calcium signaling silencing but focus on the known changes in intracellular Na^+^ homeostasis in AF and how they might affect atrial arrhythmogenesis. In addition, we summarize experimental and clinical evidence for and against I_Na,L_ inhibition as an antiarrhythmic treatment target in AF.

## 2. Atrial-Specific Intracellular Ca^2+^ Signaling

In cardiac myocytes, membrane depolarization activates voltage-gated L-type Ca^2+^ channels [24]. This initiates the influx of a small amount of Ca^2+^ into the cell. The small increase in subsarcolemmal Ca^2+^ initiates the release of bulk Ca^2+^ from the nearby Ca^2+^ storage organelle of the cardiac myocyte, the sarcoplasmic reticulum (SR), through its Ca^2+^ release channels, the ryanodine receptors type 2 (RyR2) [25,26,27]. This transient increase in the [Ca^2+^]_i_ concentration initiates the myocyte’s contraction as free Ca^2+^ binds to the myofilaments. Myocyte relaxation occurs when Ca^2+^ is released from the myofilaments, re-sequestered back into the SR by the SR Ca^2+^ ATPase, and simultaneously extruded from the cell by the sarcolemmal NCX [4,27]. This Ca^2+^-induced Ca^2+^ release (CICR) is the key mechanism underlying cardiac excitation–contraction coupling [4,25]. While fundamentally similar to ventricular myocytes, CICR in atrial cardiac myocytes is constrained by the atrial specific cyto-architecture [22,28,29,30]. Structural differences between atrial and ventricular myocytes lead to important functional differences in intracellular Ca^2+^ signaling during CICR. Specifically, atrial myocytes have a much less abundant transverse tubular (TT) system than ventricular cells [10,31,32,33,34]. TTs are invaginations of the sarcolemma that run perpendicular to the longitudinal axis of a cardiac myocyte and form a network around z-discs [35,36]. We and others have shown previously that atrial cells lack the regular TT pattern found in ventricular myocytes and often form a much less dense and more axial tubular network [10,31,32,33,37,38]. Figure 1A,B shows this difference between the TT systems in a freshly isolated living atrial and ventricular myocyte, respectively, using the membrane dye Di-8-ANEPPS. In ventricular cardiac myocytes, the sarcolemmal L-type Ca^2+^ channels and RyR2s located at the junctional SR are in close proximity throughout the cell due to the regular and dense TT network. This close apposition of L-type Ca^2+^ channels with RyR2 along the sarcolemma, the TT, and the junctional SR is called a ‘dyad’ and ensures near synchronous Ca^2+^ release throughout ventricular cardiac myocytes [4,39,40,41]. Atrial cardiac myocytes lack a regular TT network and therefore possess significantly fewer dyads. In these myocytes, Ca^2+^ release ‘propagates’ into the center of the atrial myocyte when the Ca^2+^ released through RyR2 in a dyad at the outer cell membrane activates Ca^2+^ release from a neighboring cluster of RyR2 [10,28,31]. This relatively slow process of ‘saltatory’ Ca^2+^ release from a RyR2 cluster to a neighboring RyR2 cluster results in a significant temporo-spatial delay in Ca^2+^ release between the outer membrane (sub-sarcolemmal domain) and the center of the cell in atrial myocytes [28,31,42]. Figure 1C shows a transverse confocal line scan recorded from an atrial myocyte depicting the significant delay between Ca^2+^ release at the outer membrane and the center of the myocyte. Figure 1D shows a transverse confocal line scan in a ventricular myocyte illustrating the synchronized intracellular release of bulk Ca^2+^ from the SR, the Ca^2+^ transient. It is important to note that although Ca^2+^ release in atrial myocytes is less synchronous than in ventricular myocytes, the magnitude of the Ca^2+^ released into the cytosol during steady state activation is similar in all domains of the cell [10]. Figure 2A gives a schematic overview of CICR in atrial myocytes.

### Mechanisms of Ca^2+^- and Na^+^-Based Arrhythmogenicity

When the sensitivity of the RyR2s to be activated by cytosolic Ca^2+^ increases (e.g., due to higher [Ca^2+^]_SR_ levels or increased RyR2 phosphorylation) the probability for spontaneous elemental Ca^2+^ release events (Ca^2+^ sparks) and spontaneous Ca^2+^ waves to occur, increases [43]. Under these conditions, in addition to the Ca^2+^ release triggered by an action potential (AP), there can be Ca^2+^ release events that occur ‘spontaneously’ (i.e., not triggered by an AP) during cardiac diastole. When [Ca^2+^]_i_ increases following these spontaneous Ca^2+^ release events, Ca^2+^ is extruded from the cell by the sarcolemmal NCX. Due to the electrogenicity of this transport (3 Na^+^ for 1 Ca^2+^ ion), these temporary increases in the [Ca^2+^]_i_ generate a transient inward current (I_TI_) that depolarizes the membrane potential [44]. These ‘delayed afterdepolarizations’ (DADs), which occur in diastole, can, if the depolarization threshold is reached, trigger an extrasystole and initiate arrhythmias [45,46,47]. Additionally, Ca^2+^-activated I_TI_ can occur during AP repolarization and then contribute to early afterdepolarizations (EADs), which may also trigger extrasystoles. Moreover, during short APs in the presence of high [Ca^2+^]_SR_ levels, NCX-mediated EADs can occur during the later parts of phase three of the AP [48].

In cardiac myocytes, [Ca^2+^]_i_ and [Na^+^]_i_ are tightly linked by the NCX, which uses the Na^+^ gradient across the membrane as a driving force to extrude Ca^2+^ (Figure 2B depicts the interplay between [Ca^2+^]_i_ and [Na^+^]_i_ in atrial myocytes) [49]. The NCX extrudes one Ca^2+^ ion for three incoming Na^+^ ions. In healthy cardiac myocytes during steady-state activation, NCX is the main route of Ca^2+^ extrusion [50,51] and Na^+^ entry (~60%) [18]. Thus, in cardiac myocytes, [Ca^2+^]_i_ and [Na^+^]_i_ cannot change independently of each other. In fact, the nexus of intracellular Ca^2+^ and Na^+^ homeostases is the principle behind cardiac glycoside treatment. Cardiac glycosides (e.g., ouabain or digitalis) block the Na^+^/K^+^ ATPase [52], the main route of Na^+^ extrusion in cardiac myocytes, which increases the intracellular Na^+^ concentration. High [Na^+^]_i_ levels reduce the driving force for Na^+^ entry through the NCX, which results in an increase in the [Ca^2+^]_i_ levels due to less NCX-mediated Ca^2+^ extrusion. The positive inotropic effect of cardiac glycosides is due to this intracellular Ca^2+^ ‘loading’ of the cardiac myocyte and has been used extensively in the treatment of congestive heart failure [53]. In addition, other interventions targeting the intracellular Na^+^ homeostasis in cardiac myocytes have emerged more recently. Specifically, the late component of the atrial Na^+^ current (I_Na,L_) has been identified as a treatment target in atrial fibrillation. Evidence suggests that increases in Na^+^ entry through higher I_Na,L_ levels play a role in atrial arrhythmogenesis [20,21,54,55]. In support of these data, expression in mice of a gain-of-function mutant Na_V_1.5 channel causing an increased and persistent Na^+^ current leads to the development of an atrial myopathy and spontaneous and long-lasting episodes of AF [56]. Despite its importance as a regulator of atrial cardiac myocyte function, in health and during disease, there are to date no comprehensive quantitative studies of atrial [Na^+^]_i_ regulation. For example, it is not known how atrial [Na^+^]_i_ changes during paroxysmal and persistent atrial fibrillation. More specifically, it is not known how increases in I_Na,L_ affect global [Na^+^]_i_ or whether there is an adaptive increase in Na^+^ extrusion. While many studies have examined how altered [Ca^2+^]_i_ and [Ca^2+^]_SR_ levels contribute to atrial arrhythmogenesis, the role of [Na^+^]_i_ in atrial cardiac arrhythmogenesis remains poorly understood to a large extend due to incomplete understanding of the regulation of atrial [Na^+^]_i_.

## 3. Changes in Na^+^ and Ca^2+^ Homeostasis during Atrial Fibrillation: Timing Matters

### 3.1. Changes in Intracellular Ca^2+^ and Na^+^ Signaling at the Onset of AF: Paroxysmal Atrial Fibrillation

Atrial fibrillation is a progressive disorder. Many patients go from paroxysmal to persistent AF and eventually permanent forms of the arrhythmia over the course of months or years [57,58]. Importantly, molecular and cellular changes produced by atrial fibrillation promote its chronification (‘atrial fibrillation begets atrial fibrillation’) [7]. At the same time, the rapid atrial activation rates during AF also induce ‘Ca^2+^ signaling silencing’, a complex adaptive mechanism that counteracts intracellular Ca^2+^ overload [10]. In addition, factors that precipitate the initiation of atrial fibrillation can be different from the mechanisms that sustain it.

Atrial fibrillation is characterized by high atrial depolarization rates. After the onset of the arrhythmia, the depolarization rate of the atrial myocytes increases up to tenfold [59]. This leads to a significant increase in [Na^+^]_i_ and [Ca^2+^]_i_ influx through voltage-gated Na^+^ and Ca^2+^ channels due to a sharp increase in channel openings over time. Ca^2+^ is removed from the cytosol by active transport into the SR by the SR Ca^2+^ ATPAse and by extrusion from the cell via NCX. High [Na^+^]_i_, however, reduces the driving force for Ca^2+^ extrusion via NCX by reducing the transmembrane Na^+^ gradient. This shifts Ca^2+^ removal from the cytosol toward re-sequestration into the SR, resulting in a high [Ca^2+^]_SR_. A high [Ca^2+^]_SR_ increases the frequency of arrhythmogenic DADs and EADs. In fact, experimental data from atrial myocytes isolated from patients with paroxysmal AF provide evidence for Ca^2+^-based arrhythmogenesis in paroxysmal AF. In these cells, [Ca^2+^]_SR_ levels and Ca^2+^ re-uptake into the SR were higher than in the control cells from patients in sinus rhythm. In addition, DADs were increased in these cells. Consistent with these findings, the Ca^2+^ leak from the SR was increased, as was the open probability of RyR2 [60]. In the same study, no changes in L-type Ca^2+^ current density, action potential duration, or NCX protein expression or function, all hallmarks of persistent AF, were detected [60]. This Ca^2+^-based cellular arrhythmogenicity described in atrial myocytes from patients with paroxysmal AF is consistent with the concept of high Ca^2+^ influx and cellular maladaptation during intracellular Ca^2+^ overload. When AF becomes more permanent, electrical remodeling and ‘Ca^2+^ signaling silencing’ occur (e.g., APD shortening and reduction of L-type Ca^2+^ current density [7,10,61]), which counteract the ‘Ca^2+^ flooding’ of the cell by limiting Ca^2+^ entry. For example, a recent study showed a quick reduction in L-type Ca^2+^ current density after rapid atrial pacing (7 days) due to a complex feedback loop involving high diastolic nuclear [Ca^2+^]_i_ levels and reduced levels of microRNA [62].

In addition to the high [Ca^2+^]_i_ and [Ca^2+^]_SR_ caused by rapid atrial activation during episodes of AF, certain cardiovascular comorbidities contribute to changes in intracellular Ca^2+^ homeostasis, which amplifies the development of Ca^2+^-based arrhythmogenesis. For example, in atrial myocytes isolated from dogs with systolic heart failure, Ca^2+^ release from the SR was increased, and arrhythmogenic Ca^2+^ waves were more frequent than in control cells [63]. Similarly, in a recent study on pressure overload-induced cardiomyopathy, examination of the atrial myocytes revealed an increased leak of Ca^2+^ from the SR and Ca^2+^-based arrhythmogenicity that was exacerbated when the myocytes were stretched [64]. Systolic heart failure (HFrEF) and heart failure with preserved ejection fraction (HFpEF) are risk factors for the development of AF [65]. The Ca^2+^-based arrhythmogenic mechanisms in atrial myocytes that are a product of heart failure (HFrEF or HFpEF) can likely induce an episode of paroxysmal AF. The additional Ca^2+^ ‘burden’ that is imposed on the atrial myocyte during AF amplifies the Ca^2+^-based arrhythmogenicity.

In contrast to the many studies addressing intracellular Ca^2+^ handling in atrial myocytes, relatively little is known about atrial Na^+^ homeostasis. Specifically, there are no experimental data on the regulation or behavior of [Na^+^]_i_ during paroxysmal AF. [Na^+^]_i_ levels increase in a rate-dependent manner in atrial and ventricular cardiac myocytes [10,66]. The main route for Na^+^ extrusion from the cardiac myocyte is the Na^+^/K^+^ ATPase (NKA). NKA extrudes three Na^+^ ions for two incoming K^+^ ions while hydrolyzing one ATP molecule [4,67]. The NKA is directly and tightly regulated by [Na^+^]_i_ [18,66,67,68], thus activating and augmenting the rate of Na^+^ extrusion with increasing [Na^+^]_i_ concentrations. It is, however, unclear whether atrial Na^+^ extrusion is upregulated during and after the onset of AF, where initially high [Na^+^]_i_ levels likely occur, and whether NKA expression or function are changed during paroxysmal atrial fibrillation. One recent study reported that I_Na,L_ was increased in atrial myocytes from patients with sleep-disordered breathing, which is associated with atrial arrhythmias [69]. In this study, high levels of Ca^2+^/calmodulin kinase II-mediated phosphorylation of the atrial Na^+^ channel (Na_V_1.5) led to an increase in I_Na,L_ and the frequency of cellular afterdepolarizations. This study suggests a possible role for high [Na^+^]_i_ concentrations in the onset of AF in this particular subset of patients. We have previously reported that rapid atrial pacing for 5 days leads to reduced atrial [Na^+^]_i_ in rabbit atrial myocytes, suggesting that the reduction of [Na^+^]_i_ is part of the adaptive process in response to rapid atrial activation rates [10].

### 3.2. Adaptation of Intracellular Ca^2+^ and Na^+^ Signaling during Persistent AF

After the onset of AF, molecular and cellular processes that facilitate adaptation to the high depolarization rate are initiated. Over the past 20 years, the adaptation or ‘remodeling’ of the atrial myocyte’s intracellular Ca^2+^ signaling processes to AF have been studied in many animal models of AF and in atrial myocytes isolated from patients with AF [61,70,71,72,73,74,75,76]. Remodeling of intracellular Ca^2+^ signaling during AF affects processes that limit the amount of Ca^2+^ that enters the cell and increase Ca^2+^ removal from the cytosol [28], a process known as ‘Ca^2+^ signaling silencing’ [9,10,77]. The following key features are generally found in atrial myocytes that have adapted to AF.

#### 3.2.1. Reduced L-Type Ca^2+^ Current

The L-type Ca^2+^ current (I_Ca,L_) density decreases substantially (~60–80%) during AF [71,74,78]. This is partly due to a reduction in protein expression levels of the pore-forming α_1c_ subunit. Alterations in micro RNAs appear to contribute to this reduction in the protein expression levels of the α_1c_ subunit in AF [79,80]. In addition, we and others have shown that changes in the regulation of I_Ca,L_ due to phosphatase or kinase imbalances contribute to low I_Ca,L_ levels in human AF [75,81].

#### 3.2.2. Small Ca^2+^ Transient

The intracellular Ca^2+^ transient (CaT) amplitude decreases significantly during AF [10,76]. This reduces the amount of Ca^2+^ that is available for binding to the myofilaments. The reduced CaT is therefore a significant contributor to the reduction in atrial myocyte contractility during AF [6,10,76]. We have previously shown that the reduced CaT amplitude during AF is part of the ‘calcium signaling silencing’ process that is triggered by rapid depolarization rates [9,10]. Calcium signaling silencing, which is part of the remodeling process, mitigates the effects of high [Ca^2+^]_i_ levels. Laser scanning confocal microscopy with a high spatiotemporal resolution revealed regional imbalances of CIRC during AF. While CaT amplitudes at the outer membrane in the subsarcolemmal areas of atrial myocytes were unaltered during AF, the CaT amplitude in the center of the myocyte was severely blunted, resulting in a significant reduction in the whole cell’s CaT [10].

#### 3.2.3. Upregulated Na^+^/Ca^2+^ Exchanger

The protein expression levels of NCX1 are increased during AF [76,82]. Thus, for a given cytosolic [Ca^2+^]_i_ concentration, the depolarizing current carried by NCX1 (I_NCX_) is increased during AF. When spontaneous release of Ca^2+^ from the SR occurs, the resulting higher I_TI_ increases the degree of membrane depolarization in AF atrial myocytes compared with healthy atrial myocytes [76].

#### 3.2.4. Remodeling of the RyR2 Macro-Complex

Most studies showed no changes in [Ca^2+^]_SR_ levels during AF [10,75,76], although the RyR2 complex underwent complex posttranslational modifications [60,82,83,84]. The best characterized changes are in the phosphorylation levels of two important RyR2 phosphorylation sites: Ser2808, a predominant protein kinase A (PKA) phosphorylation site, and Ser2815, a predominant Ca^2+^/calmodulin kinase II (CaMKII) phosphorylation site. Elevated phosphorylation levels at Ser2808 and Ser2815 are consistently found in AF [60,82,83,84]. Additionally, the activation of CaMKII by oxidative stress (Oxidized CaMKII) appears to contribute to increased RyR2 phosphorylation at Ser2815. The oxidized CaMKII level was increased in AF in patients [85] and was critical to AF progression in a diabetic mouse model [86]. Phosphorylation of RyR2 increases the sensitivity of the receptors to being activated by cytosolic Ca^2+^ and their open probability (p_o_) [87,88]. These changes in RyR2 Ca^2+^ sensitivity increase the propensity for the spontaneous diastolic release of Ca^2+^ from the SR independent of cell depolarization. The spontaneous release of Ca^2+^ from one or more clusters of neighboring RyR2 can further activate the release of Ca^2+^ from more adjacent RyR2 clusters, leading to a propagating intracellular Ca^2+^ wave. These Ca^2+^ waves activate NCX-mediated I_TI_ [44]. If the membrane depolarization threshold is reached, an ‘extra’ action potential can be triggered and lead to arrhythmia initiation. While this is a well-documented mechanism in ventricular arrhythmogenesis [47], it remains controversial if—and under which circumstances—arrhythmogenic spontaneous Ca^2+^ release occurs during AF. Importantly, it remains unclear whether arrhythmogenic Ca^2+^ release contributes to AF initiation and maintenance in vivo.

#### 3.2.5. Remodeling of [Na^+^]_i_ in Persistent AF

As outlined above, [Ca^2+^]_i_ and [Na^+^]_i_ cannot change independently of each other in cardiac myocytes [4,14]. While changes in intracellular Ca^2+^ signaling in AF have been studied in great detail, the regulation of intracellular Na^+^ homeostasis in healthy atrial myocytes and during AF remains poorly understood. Specifically, it is unclear how atrial remodeling in long-lasting AF affects intracellular Na^+^ homeostasis. Previous work in ventricular myocytes showed that substantive changes in [Na^+^]_i_ occur during ventricular cardiomyopathies and, specifically, that these changes are significant contributors to arrhythmogenic mechanisms in these cells. For example, in heart failure and cardiac hypertrophy, the [Na^+^]_i_ concentration is significantly increased in ventricular cardiac myocytes [17,89]. The [Na^+^]_i_ elevation under these pathological conditions seems to be related to an increase in the Na^+^ influx through voltage-gated Na^+^ channels [17] and reduced extrusion of Na^+^ by NKA [89]. In contrast to these well-documented changes in intracellular Na^+^ homeostasis in ventricular cardiomyopathies, little is known about changes in [Na^+^]_i_ during atrial fibrillation. Here, we summarize and discuss what is currently known and unknown about the adaptation of intracellular Na^+^ homeostasis during persistent AF and how it may relate to the atrial remodeling process and the chronification of AF.

Na^+^ influx: In atrial myocytes isolated from patients with atrial fibrillation, one study found that the late component of the atrial Na^+^ current (I_Na,L_) was increased, suggesting that this would lead to an increase in [Na^+^]_i_ [20,21]. A more recent study showed that I_Na,L_ in atrial myocytes from patients with AF was not different from control cells when voltage clamp measurements were performed at 37 °C [90]. An earlier study found no changes in the peak Na^+^ current (I_Na_) in atrial myocytes from patients with atrial fibrillation [91], while a study in a canine model of AF found a significant reduction in I_Na_ after 42 days of rapid atrial pacing [92]. Due to the paucity of data and the conflicting results for Na^+^ influx during persistent AF, it is unclear whether Na^+^ influx is altered during the chronification of atrial fibrillation.

Na^+^ efflux: One study examining the K^+^ dependence of the NKA current (I_NKA_) did not find any changes in atrial myocytes from AF patients compared to control cells. However, the Na^+^ dependence of I_NKA_ was not examined [93]. Experiments in this study were performed at supra-physiological intracellular Na^+^ concentrations (100 mM), where differences in I_NKA_ due to altered NKA regulation were unlikely to be detected.

Importantly, [Na^+^]_i_ has not been measured in human AF. We performed the first quantitative measurements of [Na^+^]_i_ in healthy atrial myocytes and found atrial [Na^+^]_i_ in rabbit and mouse atrial myocytes to be significantly lower than in ventricular myocytes [10,94]. It is currently unknown why [Na^+^]_i_ in healthy atrial myocytes is several mM (2–4 mM) lower than in healthy ventricular cardiac myocytes. The substantial difference between atrial and ventricular [Na^+^]_i_ suggests important differences in intracellular Na^+^ homeostasis between these two cell types. Interestingly, we also previously reported that after short-term (5 days) rapid atrial pacing, the [Na^+^]_i_ in rabbit atrial myocytes was significantly lower than in healthy atrial myocytes [10]. Thus, lowering of atrial [Na^+^]_i_ appears to be part of adaptive ‘calcium signaling silencing’ in response to rapid atrial activation rates. Our data also indicate that the intracellular Na^+^ homeostasis in atrial cells adapts differently than in ventricular myocytes when a cardiac myocyte reaches steady state, where Na^+^ influx matches the Na^+^ efflux [14,18]. For the atrial [Na^+^]_i_ level to decrease in response to rapid atrial activation, either less Na^+^ enters the cell or more Na^+^ is extruded from the cell. In large mammals, during steady state stimulation, the majority of Na^+^ enters the cell through NCX1 (~60%), and a lesser amount enters through voltage-gated Na^+^ channels (~22%). Na^+^ is extruded by the NKA (see above). Further quantitative evaluation of atrial Na^+^ influx and efflux is needed to advance our understanding of the important interplay between intracellular Na^+^ and Ca^2+^ homeostasis and its remodeling during AF.

## 4. I_Na,L_ as an Antiarrhythmic Target in Atrial Arrhythmias

During the upstroke of the AP in cardiac myocytes, voltage-gated Na^+^ channels (Na_v_) open briefly before they inactivate and enter a non-conductive state [19]. While the majority of cardiac Na_v_ remain inactivated, a small fraction of channels either does not remain inactivated, which allows them to become reactivated, or they remain open during AP repolarization [95]. This late component of the sodium current (I_Na,L_) is orders of magnitude smaller than the peak I_Na_ [20,90,95,96]. Since it was reported that the late component of I_Na_ was increased in atrial myocytes from patients with atrial fibrillation, modulation of I_Na,L_ as an anti-arrhythmic strategy in AF has been studied (Table 1) [20].

Increases in I_Na,L_ can lead to triggered activity in atrial myocytes. It is thought that reducing I_Na,L_ in patients with AF, thereby reducing the atrial [Na^+^]_i_, could prevent Ca^2+^-dependent atrial arrhythmogenesis. Ranolazine is a piperazine derivative that decreases I_Na,L_ through cardiac Na_V1.5_ channels [20,97]. Experimental studies have investigated the role of I_Na_, I_Na,L_, and their modulation in atrial arrhythmogenesis. I_Na,L_ was increased in guinea pig atrial myocytes by administration of sea anemone toxin II (ATXII), a compound known to delay the inactivation of I_Na_. In cells treated with ATXII, current clamp recordings showed a significant increase in the diastolic membrane potential, resulting in membrane depolarization. This ‘triggered’ activity was attenuated by the blocking of I_Na_ with tetrodotoxin (TTX) and ranolazine, which blocks I_Na,L_ with 40-fold higher potency than the peak I_Na_ in ventricular myocytes [98,99,100].

In a mouse model of an I_Na_ channelopathy, LQT3 (ΔKPQ), an increase in I_Na,L_ led to atrial AP prolongation and a propensity toward EADs [101]. It should be noted that prolongation of the action potentials was most prominent at slow pacing frequencies, and EADs were only observed at 0.5 Hz, which is sub-physiologic for mice, and they were never observed at higher stimulation frequencies. When SCNA5 variants from patients with long QT syndrome and early onset AF were expressed in HEK293 cells, voltage clamp experiments showed no changes in peak I_Na_ but revealed changes in channel activation and inactivation, leading to increased sustained current [102]. In isolated canine myocytes, ranolazine induced a significant prolongation of the AP only in atrial (and not in ventricular) myocytes [103]. Similarly, i.v. administration of ranolazine in anesthetized pigs resulted in a more pronounced prolongation of the effective refractory period (ERP) in the atria compared with the ventricles [104]. In a canine atrial wedge preparation, ranolazine was found to have a synergistic effect with dronedarone, a K^+^ channel blocker with effects on multiple other ion channels, in preventing the induction of AF (90% of samples) when compared with either dronedarone (17% of samples) or ranolazine (29% of samples) alone [105].

Another inhibitor of I_Na,L_, GS-458967, was developed more recently [106]. GS-458967 showed less inhibition of I_Kr_ than ranolazine [106]. In canine pulmonary vein and superior vena cava sleeve preparations, GS-458967 reduced DADs and triggered activity that was induced by isoproterenol, high calcium, or their combination [107]. GS-458967 reduced the canine atrial AP duration, decreased the AP upstroke velocity, and lengthened the ERP in a dose-dependent manner. These changes were not observed in the ventricular preparations [108]. In a porcine model of AF induced by acetylcholine and epinephrine co-administration, GS-458967 reduced AF inducibility [109]. Taken together, the experimental data suggest that when I_Na,L_ is significantly increased in healthy atrial myocytes, cellular arrhythmogenic phenomena (DADs and EADs) occur. Thus, it appears that increased I_Na,L_ might lead to atrial arrhythmogenicity, making I_Na,L_ an interesting target for anti-arrhythmic intervention in the atria. However, it should be noted that ATXII leads to 5–10-fold higher increases in I_Na,L_ than those reported for I_Na,L_ during AF [110]. In addition, a hallmark of electrical remodeling during AF is the significant shortening of the atrial AP [7,111]. This means that the effect of a moderate increase in I_Na,L_, which might occur during AF, is likely to be curtailed by the shortening of the AP. In order to better evaluate the impact of changes in I_Na,L_ on atrial arrhythmogenicity during AF, it is crucial to quantitatively evaluate the [Na^+^]_i_ and Na^+^ efflux.

Furthermore, experimental studies showed that the mechanism of action of ranolazine is not limited to I_Na,L_ and in fact inhibits several key ionic currents (I_Na_, I_Kr_, I_Ca_, I_NCX_, and I_KS_) without changes in I_to_ and I_K1_ [97]. Ranolazine has also been found to inhibit TASK-1, an atrial specific potassium channel that is upregulated in AF [112]. Given that AF at the cellular level is defined by alteration in the expression and function of key ion currents, the observed effects of ranolazine can be mediated by its effects on other currents. Ranolazine’s reduction of the upstroke velocity of the AP is mediated through its effect on the peak I_Na_. Inhibition of the peak I_Na_ is the proposed mechanism in other anti-arrhythmic agents used in the treatment of atrial fibrillation to maintain sinus rhythm, such as flecainide or propafenone [2]. Sotalol and dofetilide are two anti-arrhythmic agents used in the treatment of atrial fibrillation that exert their effect through the inhibition of I_Kr_ [113]. Ranolazine and these antiarrhythmic agents had similar potency for I_Kr_ in experimental studies [97]. The application of heterologous ion channel expression systems, such as human-induced pluripotent stem cell cardiomyocytes, in combination with novel computational methods may prove useful in further elucidating the multi ion channel blocking properties of ranolazine [114,115].

### Results of Clinical Trials Using Ranolazine in AF Treatment

Ranolazine, which had been initially indicated for the treatment of refractory angina [116], has been proposed as a novel treatment for AF. It has been suggested that the mechanism of action for ranolazine in the treatment of AF is the inhibition of I_Na,L_. As we discussed above, the role an increased atrial I_Na,L_ may play in the initiation or maintenance of AF remains controversial [20,21,90] and understudied. In addition, ranolazine efficiently inhibits many other key ion currents in cardiac myocytes, including I_Kr_, I_Ca_, and I_Na_ [97]. Thus, positive results from the utilization of ranolazine in the treatment of AF maybe unrelated to I_Na,L_.

Several studies showed that ranolazine augmented the anti-arrhythmic effect of amiodarone [117,118,119,120,121,122,123,124], an anti-arrhythmic agent commonly used in the treatment of AF [125]. These studies demonstrate that ranolazine effectively potentiates amiodarone in both the pharmacologic conversion of AF to a sinus rhythm and as a rhythm control strategy (Table 1). Amiodarone acts to prolong the action potential by inhibiting multiple currents; thus, ranolazine may act in a synergistic manner with amiodarone. Interestingly, despite the fact that ranolazine has been shown to prolong the QT interval, concomitant use with amiodarone was not shown to be pro-arrhythmic. In general, ranolazine was not shown to have significant adverse effects when compared with the control arm of studies, either with amiodarone alone or a placebo (Table 1).

The most substantial benefit for ranolazine was observed in the post-operative AF (POAF) population in the setting of open heart surgery as opposed to AF in general [120,124,129,132,133,136,137]. POAF is typically a transient event that occurs within the milieu of inflammation and adrenergic activation resulting from surgical stress [138]. In contrast to paroxysmal or persistent AF, electrical remodeling does not appear to significantly contribute to the development of POAF [139]. Thus, it is not clear what role, if any, I_Na,L_ plays in POAF or if this is the mechanism by which ranolazine exerts its effect on POAF. A recent study utilized sheep models of paroxysmal and persistent AF to examine the role of ranolazine in the cardioversion of AF to sinus rhythm. Paroxysmal and persistent atrial fibrillation were shown to have differential rotor properties. Ranolazine was found to effectively cardiovert paroxysmal AF but was ineffective in the persistent AF sheep hearts. Ranolazine may only exert an antiarrhythmic effect prior to the fibrotic changes and ion channel remodeling observed in persistent AF [126].

While the results of these studies suggest ranolazine may be useful as a stand-alone treatment of post-operative AF and in conjunction with established anti-arrhythmic agents in the conversion to and maintenance of sinus rhythm, excitement should be tempered given the relatively small patient cohorts and short follow-up times of these studies.

## 5. Outlook

The intracellular Na^+^ concentration is a potent modulator of cardiac myocyte function. It regulates the intracellular Ca^2+^ concentration, determines the activation of the Na^+^/Ca^2+^ exchanger, the Na^+^/K^+^ ATPase, and the voltage-gated Na^+^ channels. While the causal relation between a dysregulated intracellular Na^+^ homeostasis and ventricular arrhythmogenesis has been convincingly and consistently shown, it is unclear whether this same mechanism contributes to atrial arrhythmogenesis. Most importantly, quantitative measurements of atrial [Na^+^]_i_ and a more comprehensive evaluation of intracellular Na^+^ homeostasis in healthy and diseased atrial myocytes are needed in order to establish whether Na^+^-based arrhythmogenesis contributes to AF.

## Figures and Tables

**Figure 1 ijms-22-10513-f001:**
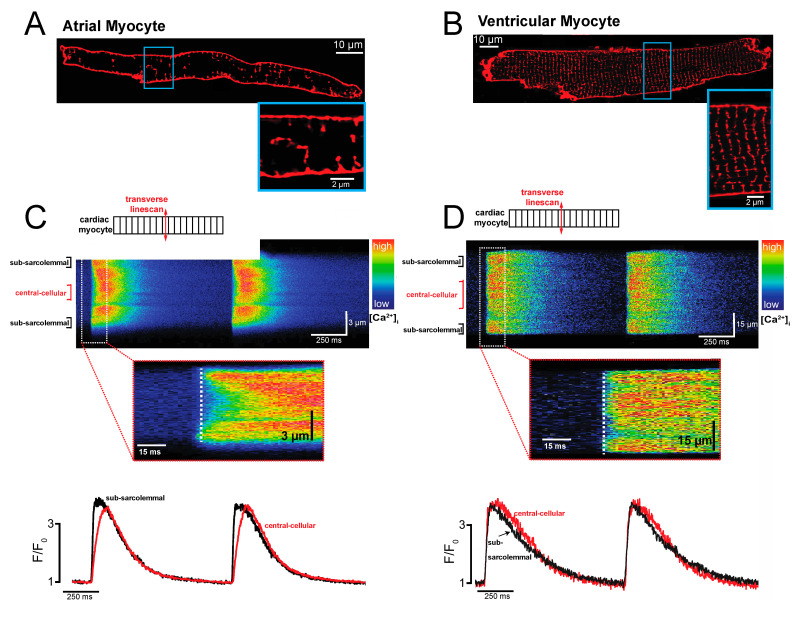
Spatiotemporal Ca^2+^ release in atrial and ventricular cardiac myocytes. (**A**) Transverse tubules. Confocal laser scanning image of a freshly isolated living atrial myocyte stained with the membrane dye Di-8-ANEPPS. (**B**) The same as in (**A**), but in a freshly isolated living ventricular myocyte. (**C**) Intracellular Ca^2+^ release. Shown is an atrial myocyte loaded with the fluorescent Ca^2+^ indicator Fluo-4. A transverse confocal linescan is used to track the spatio-temporal intracellular Ca^2+^ release from the subsarcolemmal domain to the center of the cell in an atrial myocyte during external field stimulation. The inset shows the onset of Ca^2+^ release at the outer cell membrane in the subsarcolemmal space compared to the central cellular domain. Below are the domain-specific Ca^2+^ transients derived from the confocal images shown in the upper panel. (**D**) The same as in (**C**) but for a ventricular myocyte.

**Figure 2 ijms-22-10513-f002:**
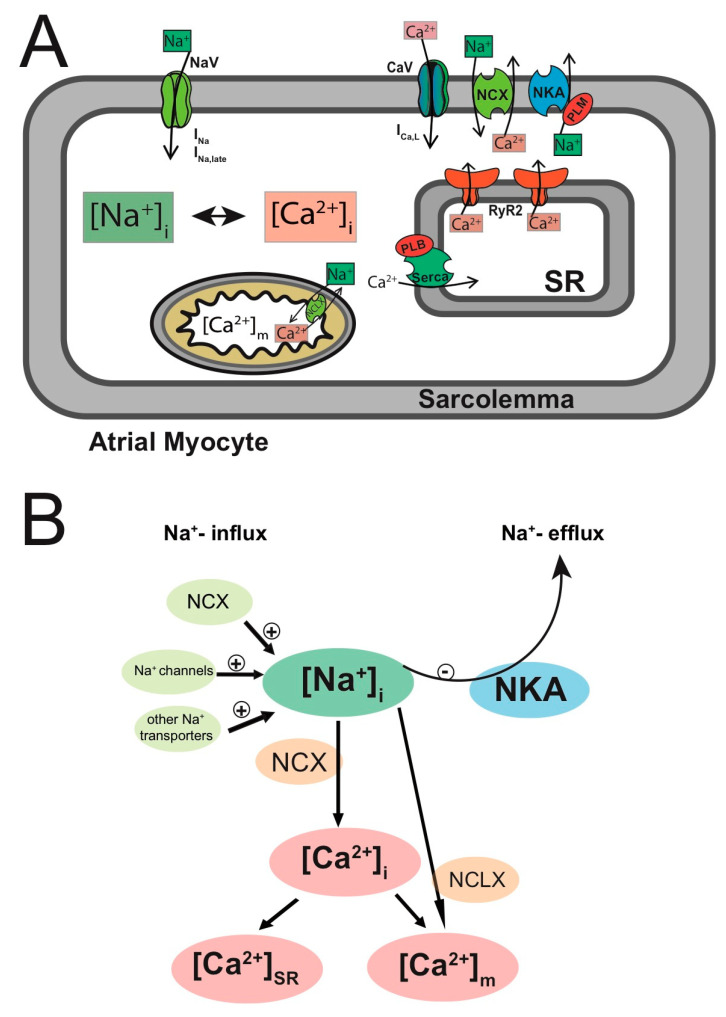
Intracellular Na^+^ and Ca^2+^ signaling in atrial myocytes. (**A**) Schematic depicting intracellular Na^+^ and Ca^2+^ signaling in atrial myocytes. (**B**) Schematic depicting the interdependence of intracellular Ca^2+^ and Na^+^ homeostases. See text for details. [Na^+^]_i_: intracellular Na^+^ concentration; [Ca^2+^]_i_: intracellular Ca^2+^ concentration; NaV: voltage-dependent Na^+^ channels; I_Na_: Na^+^ current; I_Na,late_: ‘late’ Na^+^ current; CaV: L-type Ca^2+^ channel (CaV1.2); I_Ca, L_: L-type Ca^2+^ current; NCX: Na^+^/Ca^2+^ exchanger; I_NCX_: Na^+^/Ca^2+^ exchange current; NCLX: mitochondrial Na^+^/Ca^2+^ exchanger; NKA: Na^+^/K^+^ ATPase; I_NKA_: Na^+^/K^+^ ATPase current; PLM: phospholemman; RyR2: ryanodine receptor type 2; Serca: sarcoplasmic reticulum Ca^2+^ ATPase; PLB: phospholamban; SR: sarcoplasmic reticulum; [Ca^2+^]_SR_: SR Ca^2+^ concentration; [Ca^2+^]_m_: mitochondrial Ca^2+^ concentration.

**Table 1 ijms-22-10513-t001:** Clinical studies testing ranolazine for AF treatment.

Study	Population Studied (N, Age, %Male)	Study Design	AF Detection Method, Surveillance Duration	Results
Murdock 2008 [126] ^#^	Recurrent AF with failure to AF ablation or anti-arrhythmic behavior 7, 67 ± 9, 57%	Oral RN (500–1000 mg/BID) after stopping all other anti-arrhythmic therapy	Not reported	AF conversion: 5/7 AF recurrence: 4/7 remained in NSR 1 AF event at 3 months and 6 months
Miles 2011 [127] ^a^	Post CABG AF 182, 66.7 ± 9.3, 70% (intervention arm) 211 64.9 ± 10.9, 77% (control arm)	Intervention arm: 1500 mg RN before surgery, 1000 mg RN BID post-op for 10–14 days Control arm: 400 mg amiodarone before surgery, 200 mg amiodarone BID post-op 10–14 days	Continuous ECG monitoring throughout hospitalization	Incidence of POAF: AF 26.5% in control group vs. 17.5% in RN-treated group (*p* = 0.035)
Fragakis 2012 [117] ^c^	New onset AF (<48 h from diagnosis) 25, 62 ± 8, 60% (intervention arm) 26, 64 ± 7, 69% (control arm)	Intervention arm:1500 mg RN daily and IV amiodarone Control arm: IV amiodarone (loading dose: 5 mg/kg in 1 h followed by 50 mg/h for 24 h or until cardioversion)	Continuous ECG in CCU for 24 h followed by >1 day inpatient	Conversion rate to NSR: 65% in control vs. 88% in RN-treated group (*p* = 0.056) Time to conversion: control 14.6 ± 5.3 h vs. RN-treated 9.8 ± 4.1 h (*p* < 0.001)
Murdock 2012 [128] ^#^	Recurrent AF with electro-cardioversion failure 25, 62 ± 11, 76%	2000 mg RN given after failed electrocardioversion attempt, repeat electrocardioversion after 3–4 h of administration	Not reported	AF conversion: 17/25 3 patients spontaneously converted before the second attempt at EC within 4 h of ranolazine
Tagarakis 2013 [129] ^c^	Post-CABG AF 34, NA, NA (intervention arm) 68, NA, NA (control arm)	Intervention arm: 375 mg RN BID 3 days prior to operation until discharge Control arm: usual care	Continuous ECG monitoring for first 24 h followed by ECG monitoring every 4 h until discharge	Incidence of POAF: control 30.8% vs. RN-treated 8.8% (*p* < 0.001)
Koskinas 2014 [118] ^c^	New onset AF (<48 h from diagnosis) 61, 66 ± 11, 41% (intervention arm) 60, 64 ± 9, 48% (control arm)	Intervention arm: 1500 mg RN daily and IV amiodarone Control: IV amiodarone (loading dose: 5 mg/kg in 1 h followed by 50 mg/h)	Continuous ECG monitoring in the CCU for 24 h	Conversion rate at 12 h: control 32% vs. RN-treated 52% (*p* = 0.021) Conversion rate at 24 h: control 70% vs. RN-treated 87% (*p* = 0.024) Time to conversion: control 13.3 ± 4.1 h vs. RN-treated 10.2 ± 3.3 h (*p* < 0.001) Modest QT prolongation in both the groups, no serious adverse reactions, and no pro-arrhythmic events.
Simopuolos 2014 [119] ^c^	Post-CABG AF 20, 69 ± 7, 70% (intervention arm) 21, 67 ± 8, 60% (control arm)	Intervention arm: 500 mg RN (loading dose) followed by 375 mg RN BID and IV amiodarone Control arm: IV amiodarone: 300 mg in 30 min followed by 750 mg in 24 h, then 200 mg BID for one week and then 200 mg daily for 1 week	Continuous ECG monitoring for first 24 h followed by ECG every 4 h, monitoring until discharge	Time to conversion to NSR: control 37.2 ± 3.9 h vs. RN-treated 19.6 ± 3.2 h (*p* < 0.001)
Scirica 2015 (MERLIN) [130] ^c^	Patients hospitalized for NSTEMI 3162, 17% >75 yrs, 66.8% (intervention arm) 3189, 18% >75 yrs, 63.7% (control arm)	Intervention arm: IV 200 mg RN with 80 mg/h infusion for 12–96 h, then 1000 mg oral RN BID Control arm: placebo plus standard medical intervention	Continuous ECG monitoring for 7 days Median clinical follow-up at 12 months	AF burden—episodes detected on continuous ECG in first 7 days: control 55 (1.7%) vs. RN-treated 75 (2.4%) (*p* = 0.08) New onset AF: Clinically insignificant: control 7 vs. RN-treated 5 Paroxysmal: control 48 vs. RN-treated 18 Chronic: control 20 vs. RN-treated 28 (*p* < 0.01) One-year AF events: control 4.1% vs. RN-treated 2.9% (*p* = 0.01)
De Ferrari 2015 (RAFFAELLO) [131] ^c^	Persistent AF, 2 h after successful cardioversion 65, 66.9 ± 11.8, 70.8% (375 mg RN) 60, 65.5 ± 8.5, 85% (500 mg RN) 58, 63.6 ± 11.3,79.3 (750 mg RN) 55, 65.2 ± 9.5, 74.5% (control arm)	Intervention arm: either oral 375 mg BID, 500 mg BID, or 750 mg BID ranolazine Control arm: placebo	Transtelephonic electrocardiogram for 16 weeks and 12 lead ECG at 1 week, 2 months, and 4 months	AF recurrence: control 56.4% vs. 375 mg (56.9%) vs. 500 mg (41.7%) vs. 750 mg (39.7%) AF in higher dose vs. control (*p* = 0.053)
Tsanaxidis 2015 [120] *^,c^	New onset AF 36, 67 ± 10,25% (intervention arm) 29, 62 ± 11,55% (control arm)	Intervention arm: 1000 mg RN once + IV amiodarone Control arm: IV amiodarone (loading dose: 5 mg/kg in 1 h followed by 50 mg/h)	Not reported	Time to conversion to NSR: control 24.4 ± 4.1 vs. 8.1 ± 2.2
Bekeith 2015 [132] *^,b^	POAF 27, NA, NA (intervention arm) 27, NA, NA (control arm)	Intervention arm: 1000 mg RN BID for 48 h prior to surgery and 2 weeks post-op Control: placebo	ECG monitoring in patient followed by holter monitor 2 weeks post-discharge	Incidence of AF: control 8 (30%) vs. RN-treated 5 (19%) (*p* = 0.530)
Hammond 2015 [133] ^a^	POAF 69, 59.7 ± 10.8, 68.1% (intervention arm) 136, 62.2 ± 11.8, 56.6% (control arm)	Intervention arm: 1000 mg RN BID starting on day of surgery for 7 days or until discharge Control arm: standard therapy	Not reported	POAF occurrence: 41.9% vs. 10.1% (*p* < 0.0001)
Reiffel 2015 (HARMONY) [121] ^a^	Paroxysmal AF with recent dual-chamber pacemaker placement 26, 70 ± 10.8, 39% (intervention arm) 52, 73.5 ± 11.5, 44.5% (control arm)	Intervention arm: 750 mg RN BID, dronedarone, or both Control: placebo	Dual-chamber pacemaker, 4-week run-in period followed by a 12-week treatment period	AF burden: % difference vs. placebo dronedarone 9% (*p* = 0.78), RN −20% (*p* = 0.49), RN + 150 mg dronedarone −43% (*p* = 0.072), RN + 225 mg dronedarone (*p* = 0.008)
Tsanaxidis 2017 [122]	New onset AF (<48 h from onset) 92, 70 ± 10, 41% (intervention arm) 81, 67 ± 11, 50.6% (control arm)	Intervention arm: 1000 mg RN once and IV amiodarone Control arm: IV amiodarone (loading dose: 5 mg/kg in 1 h followed by 50 mg/h)	Not reported	Time to conversion: control 19.4 ± 4.4 vs. RN-treated 8.6 ± 2.8 (*p* < 0.0001) Conversion rate at 24 h: control 58% vs. RN-treated 98% (*p* < 0.001)
Simopoulos 2018 [123]	POAF in patients with HFrEF vs. HFpEF 511, 65 ± 9, 87% (HFrEF arm) 301, 66 ± 10, 85% (HFpEF arm)	Intervention arm: 500 mg RN followed by 375 mg RN after 6 h and 375 mg RN BID thereafter and amiodarone Control arm: IV amiodarone (300 mg in first 30 min + 1125 mg over next 36 h)	Not reported	Time to conversion: HFrEF: control 24.3 ± 4.6 vs. RN-treated 10.4 ± 4.5 HFpEF: control 26.8 ± 2.8 vs. RN-treated 12.2 ± 1.1
**Meta-Analysis**				
Guerra 2017 [134]				AF event (new onset or recurrence): OR 0.47; 95% CI 0.29–0.76 (*p* = 0.003) POAF: OR 0.29; 95% CI 0.11–0.77 (*p* = 0.03) Non-operative AF: OR 0.70; 95% CI 0.54–0.83 (*p* = 0.005) Successful cardioversion vs. amiodarone alone: OR 3.11; 95% CI 1.42–6.79 (*p* = 0.004) Time to cardioversion: SMD −2.83 h; 95% CI from −4.69 to −0.97 h (*p* < 0.001)
Gong 2017 [135]				AF event: RR 0.67, 95% CI 0.62–0.87 (*p* = 0.002) Successful cardioversion vs. amiodarone alone: RR 1.23, 95% CI 1.08–1.40 (*p* = 0.002) Time to cardioversion: WMD –10.38 h; 95% CI from −18.18 to −2.57 h (*p* < 0.009)

AF = atrial fibrillation, BID = twice daily, CABG = coronary artery bypass surgery, HFpEF = heart failure with preserved ejection fraction, HFrEF = heart failure with reduced ejection fraction, NSR = normal sinus rhythm, NSTEMI = non-ST elevation myocardial infarction, POAF = post-operative AF, RN = ranolazine. ^#^ Case series. * Abstract. ^a^ Included in Guerra et al. meta-analysis. ^b^ Included in Gong et al. meta-analysis. ^c^ Included in both Guerra et al. and Gong et al. meta-analyses.

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
