# Peer review of "Calcium Signaling Silencing in Atrial Fibrillation: Implications for Atrial Sodium Homeostasis"

_ijms, 2021, doi:10.3390/ijms221910513_

Round 1

Reviewer 1 Report

In the review, the authors summarized the current progress of the role of Ca2+ and Na+ handling in atrial fibrillation (AF). The authors started with a general overview of Ca2+ and Na+ homeostasis and regulation of intracellular Ca2+ and Na+. Then the authors focused on the dysregulation of Ca2+ handling and contribution to AF. Last, the authors summarized the clinical studies by targeting late Na+ current. Overall, the manuscript is well written. But I do have some comments.

Major points:

  1. The review is well written, but generally, the review lacks deep discussion, instead of a summary of previous studies. Particularly, in some sections, there were only a few sentences. The authors should combine them or expand the sections with a deeper discussion.
  2. AF has different types, typically paroxysmal and chronic AF. It is better to the discussion of the role of Ca2+ and Na+ handling in different types of AF, and also on the initiation and maintenance of each type of AF.
  3. The manuscript focused on the Ca2+ and Na+ concentration and current. It is also necessary to discuss the role of ion channel expression and post-translational modification such as the role of CaMKII in regulating Ca2+ and Na+ current and onset of AF.
  4. The authors should include more recent studies of Ca2+ handling in AF. For example, one recent study by Stanley Nattel’s group described the role of IP3 receptor and nuclear Ca2+ in AF (Qi et al., Cir res, 2020).

Minor points:

  1. It is a bit bizarre to describe Figure 1 B before Figure 1A. The authors should change the panel order in Figure 1.
  2. Line 102 “Fig.1A shows” should be panel C. and paned D was not mentioned in the text.

Author Response

Responses to reviewers:

We would like to thank the reviewers for the insightful comments on our manuscript entitled "Calcium Signaling Silencing in Atrial Fibrillation: Implications for Atrial Sodium Homeostasis". We have carefully revised the manuscript and have addressed all comments raised by the reviewers. The point-by-point responses are below:

Reviewer 1

We would like to thank the reviewer for the careful reading of our manuscript and the specific comments, which we have addressed in the revised version of the manuscript.

Major comments:

  1.  "The review is well written, but generally, the review lacks deep discussion, instead of a summary of previous studies. Particularly, in some sections, there were only a few sentences. The authors should combine them or expand the sections with a deeper discussion."

Authors' response: We have expanded our discussion significantly to address this issue. Specifically, we have added to sections 2. and 3. in the revised version of the manuscript (see ll 107-572). We would like to stress that the focus of this review, and its novelty, lies in the discussion of the atrial Na+ homeostasis, its role during atrial fibrillation and the potential role of the late Na+ current as an anti-arrhythmic target (as stated in the introduction).

  1. "AF has different types, typically paroxysmal and chronic AF. It is better to the discussion of the role of Ca2+ and Na+ handling in different types of AF, and also on the initiation and maintenance of each type of AF."

Authors' response: We have implemented this in the revised version of the manuscript. We have now added a section on paroxysmal and persistent AF (see ll 339 ff and ll 435 ff).

  1. "The manuscript focused on the Ca2+ and Na+ concentration and current. It is also necessary to discuss the role of ion channel expression and post-translational modification such as the role of CaMKII in regulating Ca2+ and Na+ current and onset of AF."

Authors' response: We have included this in the expansion and revision of sections 2. and 3. of the revised versions of the manuscript (see ll 339 ff and ll 435 ff).

  1. "The authors should include more recent studies of Ca2+ handling in AF. For example, one recent   study by Stanley Nattel’s group described the role of IP3 receptor and nuclear Ca2+ in    AF (Qi et           al., Cir res, 2020)."

      Authors' response: We have added this study to the revised version of the manuscript (ll 360-363).

Minor comments:

"1. It is a bit bizarre to describe Figure 1 B before Figure 1A. The authors should change the panel order in Figure 1"

Authors' response: We have slightly changed this figure by showing exemplars of live cell imaging of t-tubules using Di-8-Anepps. We have reconfigured the text so that Figure 1A is now introduced before Figure 1 B (ll78-80).

  1. "Line 102 “Fig.1A shows” should be panel C. and paned D was not mentioned in the text."

Authors' response: We have changed this in the revised version of the manuscript. ("Figure 1C shows a transverse confocal line scan recorded from an atrial myocyte depicting the significant delay between Ca2+ release at the outer membrane and the center of the myocyte. Figure 1D shows a transverse confocal line scan in a ventricular myocyte illustrating the synchronized intracellular release of bulk Ca2+ from the SR, the Ca2+ transient. ", ll 100-103).

Reviewer 2 Report

The paper entitled 'Calcium Signaling Silencing in Atrial Fibrillation: Implications for Atrial Sodium Homeostasis' by Kaplan et al. is reviewing the regulatory mechanisms of Na+ homeostasis in atrial arrhythmogenesis. Clinical studies testing ranolazine for AF treatment are also reviewed.

The paper is carefully written and reviews the most important mechanisms of changes in Na+ and Ca2+ homeostasis associated with atrial fibrillation, including changes at the onset or during AF, changes in Ca2+ transients amplitude, reduction of L-type Ca2+-current, upregulation of the Na+/Ca2+ exchanger or posttranslational modifications of RyR2 complex.

I recommend minor changes to improve the paper

  1. I consider that the use of 'silencing' in the title is misleading, as it is commonly used as gene silencing. Instead, clinical studies in AF are dedicated to pharmacological blockade and not to genetic engineering. Therefore I recommend the replacement of silencing with 'blockade' or 'inhibition'.
  2. Section 2. I recommend the authors to consider the paper of Mann et al. 2019 (PMID 31228558) that considers the analyzes of multiple ion currents components (e.g. voltage-gated Na+ current, L-type Ca2+ current, transient outward K+ current, delayed rectifier K+ current, and "funny" hyperpolarization-activated current) to evaluate proarrhythmogenic liability of drug candidates.  A recent review published by Amuzescu et al. 2021 (PMID 33607174) also considers the most relevant mathematical models employed to study various arrhythmogenesis mechanisms and would be very useful to be included in your paper.
  3. Section 3. subsection Results of Clinical Trials using Ranolazine in AF Treatment. I recommend the inclusion of the studies Ramirez et al. 2019 (PMID 31594392), Patel and Kluger 2018 (PMID 30009098), De Vecchis et al. 2018 (PMID De Vecchis), Ratte et al. 2019 (PMID 32038227).

Author Response

Responses to reviewers:

We would like to thank the reviewers for the insightful comments on our manuscript entitled "Calcium Signaling Silencing in Atrial Fibrillation: Implications for Atrial Sodium Homeostasis". We have carefully revised the manuscript and have addressed all comments raised by the reviewers. The point-by-point responses are below:

Reviewer 2

We would like to thank the reviewer for the favorable and encouraging comments on our manuscript. We have addressed all specific comments in the revised version of the manuscript.

Reviewer comments:

  1. "I consider that the use of 'silencing' in the title is misleading, as it is commonly used as gene silencing. Instead, clinical studies in AF are dedicated to pharmacological blockade and not to genetic engineering. Therefore I recommend the replacement of silencing with 'blockade' or 'inhibition'."

Authors' response: We thank the reviewer for this comment. While we agree that 'silencing' can refer to gene silencing we have previously discovered and described a complex cellular and molecular adaptive process in atrial myocytes as a response to rapid rate as "Ca2+ signaling silencing" (Greiser et al, " Tachycardia-induced silencing of subcellular Ca2+ signaling in atrial myocytes." The Journal of Clinical Investigation, 2014, 124:4759-72 and Greiser " Calcium signalling silencing in atrial fibrillation." J Physiol 2017, 595:4009-4017). As the 'silencing' in our title refers to this specific term, we would like to leave the title as is.

  1. 2. "Section 2. I recommend the authors to consider the paper of Mann et al. 2019 (PMID 31228558) that considers the analyzes of multiple ion currents components (e.g. voltage-gated Na+current, L-type Ca2+current, transient outward K+ current, delayed rectifier K+ current, and "funny" hyperpolarization-activated current) to evaluate proarrhythmogenic liability of drug candidates.  A recent review published by Amuzescu et al. 2021 (PMID 33607174) also considers the most relevant mathematical models employed to study various arrhythmogenesis mechanisms and would be very useful to be included in your paper."

Authors' response: We have added these studies to the revised version of the manuscript ("The application of heterologous ion channel expression systems, such as human induced pluripotent stem cell cardiomyocytes, in combination with novel computational methods may prove useful in further elucidating the multi ion-channel blocking properties of ranolazine." ll 624-627).

  1. " Results of Clinical Trials using Ranolazine in AF Treatment. I recommend the inclusion of the studies Ramirez et al. 2019 (PMID 31594392), Patel and Kluger 2018 (PMID 30009098), De Vecchis et al. 2018 (PMID De Vecchis), Ratte et al. 2019 (PMID 32038227)."

Authors' response: We have included the suggested studies in the revised version of the manuscript (Ramirez et al ll 648-653; Patel et al l 644; de Vecchis et al l 578 and Ratte et al ll 617-618).